# A Singular and Widespread Group of Mobile Genetic Elements: RNA Circles with Autocatalytic Ribozymes

**DOI:** 10.3390/cells9122555

**Published:** 2020-11-28

**Authors:** Marcos de la Peña, Raquel Ceprián, Amelia Cervera

**Affiliations:** IBMCP (CSIC-UPV), Ingeniero Fausto Elio s/n, 46022 Valencia, Spain; rceprian@uoc.edu (R.C.); amcerol@ibmcp.upv.es (A.C.)

**Keywords:** circular RNA, retrotransposons, ribozyme

## Abstract

Circular DNAs, such as most prokaryotic and phage genomes, are a frequent form of nucleic acids, whereas circular RNAs had been regarded as unusual macromolecules until very recently. The first reported RNA circles were the family of small infectious genomes of viroids and circular RNA (circRNA) satellites of plant viruses, some of which contain small self-cleaving RNA motifs, such as the hammerhead (HHR) and hairpin ribozymes. A similar infectious circRNA, the unique human hepatitis delta virus (HDV), is another viral satellite that also encodes self-cleaving motifs called HDV ribozymes. Very recently, different animals have been reported to contain HDV-like circRNAs with typical HDV ribozymes, but also conserved HHR motifs, as we describe here. On the other hand, eukaryotic and prokaryotic genomes encode sequences able to self-excise as circRNAs, like the autocatalytic Group I and II introns, which are widespread genomic mobile elements. In the 1990s, the first circRNAs encoded in a mammalian genome were anecdotally reported, but their abundance and importance have not been unveiled until recently. These gene-encoded circRNAs are produced by events of alternative splicing in a process generally known as backsplicing. However, we have found a second natural pathway of circRNA expression conserved in numerous plant and animal genomes, which efficiently promotes the accumulation of small non-coding RNA circles through the participation of HHRs. Most of these genome-encoded circRNAs with HHRs are the transposition intermediates of a novel family of non-autonomous retrotransposons called retrozymes, with intriguing potential as new forms of gene regulation.

## 1. Introduction

Covalently closed circular RNA molecules (circRNAs) have been historically regarded as a peculiar form of nucleic acids. However, circular DNAs, such as plasmids or many bacteriophage, viral, plastid and prokaryotic genomes, are ubiquitous molecules among all areas of life. This view has changed dramatically in recent years with the discovery of new examples of circRNAs of diverse origin, with intriguing regulatory and biotechnological capabilities. Now, we know the existence of several types of circRNAs, such as infectious agents with circRNA genomes, self-excising Group I and II introns, the circRNA intermediates of non-autonomous retrotransposons with ribozymes (retrozymes) or the intronic/exonic circRNAs [1]. It is somehow surprising that, despite their original discovery more than 40 years ago [2], the abundance and roles of circRNAs in biology have remained poorly studied. Probably, their discovery as an atypical family of minimal subviral genomes, together with the unique biochemical and biophysical features of circRNAs compared with classic linear RNAs, had hindered the interest in the study of RNA circles until recently. We recognize that circRNAs are frequent in most cell transcriptomes, showing differential properties that cannot be found in linear RNAs, such as a high stability against exonucleases, a more definite fold, or their capability for an efficient replication by a rolling circle mechanism.

## 2. The Discovery of the First CircRNAs

The first circRNAs were discovered in the 1970s as a group of minimal infectious agents named viroids and viroid-like circRNA satellites [2,3]. These subviral organisms infect some flowering plants, either autonomously (viroids), or in the presence of a helper virus (virus satellite circRNAs). There is even an exceptional example of a viroid-like circRNA integrated in the genome of carnation plants [4,5], indicating the heterogeneity of this group of mobile genetic elements. They all have a small (220–400 nt) non-protein-coding genome of circular RNA [6]. Except for the members of one of the viroid families (*Pospiviroidae*), these circRNAs encode small self-cleaving RNA motifs or ribozymes in either one or both of the genomic polarities (Figure 1A) [7].

These nucleolytic ribozymes, such as the hammerhead (HHR) and the hairpin (HPR) ribozymes (see below), catalyze a transterification reaction in a specific point of the RNA sequence, which promotes the cleavage of the multimeric copies during their rolling circle RNA replication. A somewhat similar infectious agent with a circRNA genome is a satellite RNA associated with the human hepatitis B virus (HBV), the so-called hepatitis delta virus (HDV) (Figure 1B) [8]. The circRNA genome of the HDV also encodes small self-cleaving ribozymes in both RNA polarities (the HDV ribozymes), required for the processing of the multimeric RNAs during HDV replication. A particular feature of the larger circRNA genome of the HDV (~1680 nt) is the presence of an open reading frame in one of the polarities (the so called delta antigen) [9], an exceptional capability among most infectious circRNAs of plants [10]. The HDV agent has only been detected in humans so far, and it has been, for decades, the sole representative of the genus *Deltavirus*. Very recently, however, the presence of divergent HDV-like circRNAs have been detected in the transcriptomes of diverse animals, both invertebrates and vertebrates, including mammals [11,12,13,14]. Remarkably, no hepadnavirus co-infection has been detected for any of these novel HDV-like RNAs, although the participation of other helper viruses in the lifecycle of these circRNAs cannot be ruled out [15]. The HDV-like circRNAs of amniotes such as birds [11], snakes [12], and a rodent [13] contain the characteristic HDV ribozyme motifs, which show a similar sequence and genomic location to the ribozymes present in the human HDV deltavirus (Figure 1B). However, HDV-like circRNAs reported in other unrelated metazoans, such as termites or amphibians [14], do not seem to contain sequences compatible with HDV ribozymes. Following some structure-based bioinformatic searches, we have detected the conserved presence of HHR sequences in both polarities of the HDV-like circRNAs of toads and termites (Figure 1B) (de la Peña et al., in preparation). These HHRs are novel type III variants, which share intriguing similarities, such as longer than usual helixes or conserved pseudoknot interactions (Figure 1B).

Altogether, and with the sole exception of plant *Pospiviroids*, the heterogeneous group of infectious circRNAs shares the presence of small self-cleaving motifs (either HHR, HPR or HDV ribozymes that can somehow be regarded as interchangeable motifs) as a common key feature among these mobile genetic elements. These data, together with the discovery of genome-encoded circRNAs with ribozymes in eukaryotes, open a more complex scenario than that previously imagined for the family of infectious circRNAs such as viroid-like and HDV-like agents. In this regard, the classical association of HDV with HBV would likely be specific to humans, whereas new metazoan HDV-like agents could be either examples of viral satellite RNAs or even a novel family of autonomous viroid-like agents in animals.

Among other open questions regarding these minimal infectious agents, an especially intriguing one relates to their evolutionary origins. Historically, they were reported almost 50 years ago as minimal replicating entities at the frontiers of life. This uniqueness, followed by the hypothesis of ribozymes as the most ancient biocatalysts, fueled the theory that infectious circRNAs could be “living fossils” from the prebiotic RNA world [16,17,18,19]. However, the absence of a reasonable evolutionary path that could only account for the presence of these putative RNA fossils only in a few eukaryotes (either angiosperms or metazoans), clashes with the assumption of these circRNAs as direct survivors of the prebiotic RNA world. Moreover, we know now that small self-cleaving RNA motifs are widespread in all domains of life, from bacteriophages to humans [7,20], indicating that the presence of nucleolytic ribozymes in many subviral RNA agents is in fact a very frequent phenomenon in DNA genomes. These observations, together with the discovery of novel circRNAs with ribozymes in disparate organisms, reinforce a more parsimonious hypothesis, where infectious circRNAs in eukaryotes could have emerged de novo several times during evolution from the more frequent genome-encoded circRNAs with ribozymes [21].

## 3. Genome-Encoded CircRNAs

The first example of a natural circRNA encoded in a DNA genome was the intervening sequence of the rRNA precursor of *Tetrahymena thermophila* [22]. The characterization of the splicing mechanism of this intervening sequence allowed the discovery of the first catalytic RNA: the self-splicing Group I intron [23]. Group I introns are widespread mobile genetic elements that follow a patchy distribution among bacteria, eukaryotes, and viruses and phages [24]. Although the circular RNA forms of Group I introns still lack a clear functional significance, the formation of full-length intron circles is a general feature of this family of transposable elements [25]. Another family of self-splicing ribozymes capable of generating RNA circles in vivo is the Group II introns. They are also transposable elements present in some bacteria and eukaryotic organelles, which show a small proportion of circRNAs during their splicing, without any clear role [26,27].

In 1993, the first examples of circRNAs derived from protein-coding genes were reported in mammals [28,29], which contributed to the understanding of previous observations on “scrambled exons” [30,31], although these discoveries were regarded as anecdotical. Twenty years later, different labs found out that numerous splicing-derived circRNAs occur in most eukaryotes [32,33,34]. These circRNAs are expressed in different tissues and at different levels, which in some cases are much higher than the main linear transcript [33]. Most gene-encoded circRNAs come from pre-mRNA splicing through the spliceosome, wherein a downstream 5′- splice site is joined to an upstream 3′- splice site through a process known as “backsplicing” [35]. Among the diverse biological functions proposed for these circRNAs, the regulation of splicing [36] and transcription [37], small RNAs biology [38] or even RNA-mediated inheritance [39] have all been suggested (for a review see [40]).

There is a second natural pathway of circRNA production in eukaryotes. In our lab, we have reported the expression of abundant circular RNAs encoded in the genomes of plants [21,41] and metazoans [42], which does not require the splicing machinery of the cell. These genome-encoded RNA circles all contain small self-cleaving motifs of the HHR family [20], which occur in tandem copies capable of self-processing the RNA during its transcription (Figure 2).

## 4. Autocatalytic RNA Cleavage in Diverse RNA Circles

More than 30 years ago, the laboratories of Tom Cech and Sydney Altman reported the ground-breaking discovery of catalytic RNAs or ribozymes [23,43]. This change of paradigm boosted the field of RNA biology, which has resulted in numerous discoveries regarding the roles and capabilities of this macromolecule. The finding of natural ribozymes also reinforced the hypothesis of the prebiotic RNA world [44], where the RNA molecule carried out both catalytic (ribozymes) and genetic (RNA genomes) roles. This hypothesis is strongly supported by the presence of RNA genomes among the simplest organisms (RNA viruses, RNA satellites and viroids), but also by the existence of key ribozymes such as the ribosome, the central machine of life [45], which catalyzes the peptide bond formation in all living organisms.

Among natural ribozymes, the family of small self-cleaving RNAs (50–200 nt) is likely the most enigmatic group of catalytic RNAs. These motifs catalyze a sequence-specific intramolecular transesterification that promotes the breakage (or ligation) of the RNA chain. There are so far nine different classes of small self-cleaving ribozymes described, such as the HHR [46,47], HPR [48], HDV [49], Varkud-satellite (VS) [50], GlmS [51], twister [52], twister sister, hatchet and pistol [53] ribozymes. The HHR is probably the most studied and frequent small ribozyme. It is composed of a conserved catalytic core of 15 nucleotides surrounded by three helixes (I to III). The whole motif adopts a γ-shaped fold where helix I usually interacts with helix II through loop–loop interactions required for efficient in vivo activity [54,55,56]. Three circularly permuted topologies are possible for the HHR, named type I, II or III, depending on the open-ended helix (Figure 3). As mentioned above, the HHR was originally discovered in the circRNA genomes of plant viral RNA satellites and viroids [46,47], but then, they were also reported in the genomes of unrelated eukaryotes such as newts, trematodes, plants, and some mammals [57,58,59,60,61].

More recently, different labs have reported that HHR motifs are widespread from prokaryotic to eukaryotic genomes [62,63,64,65], including the human genome [20,66,67,68]. Type I HHR motifs are typically found in metazoan genomes, type III HHR motifs mostly occur in plants and in the circRNA agents that infect them, whereas prokaryotic and bacteriophage genomes show the presence of either type I, II or III HHRs (Figure 1). A similar widespread occurrence of genomic HHRs has been reported for two other catalytic RNAs such as the HDV [69,70] and twister ribozymes [52]. The biological roles of most of these genomic self-cleaving RNAs seem to be involved in the biology of retrotransposons and other mobile genetic elements. In this way, HDV-like ribozymes have been reported as conserved motifs in the family of R2 retrotransposons [71] and other LINE retroelements [72,73]. Similarly, we have recently detected that truncated but catalytically competent versions of the twister ribozyme are conserved in the 5′- end of retrotransposons of the RTE superfamily (Martin and de la Peña, in preparation). The most likely role for all these self-cleaving motifs would be to carry out proper 5′-end processing of the retrotransposon RNA from upstream transcripts, but also to promote translation initiation [72]. On the other hand, the families of Penelope-Like Elements (PLEs) [74] and Terminon retrotransposons [75] show the conserved presence of type I HHR motifs, although they correspond to minimal variants of this ribozyme lacking the helix III or the characteristic tertiary loop–loop interactions, and yet with an unknown role.

Finally, classical type I and III HHRs (Figure 3), abundant in animal and plant genomes, respectively, have been involved in the processing of a novel family of non-autonomous retrotransposons, the so-called retrozymes—for retrotransposons with hammerhead ribozymes, which spread through small circRNA transposition intermediates (Figure 2).

## 5. Hammerhead Ribozymes in the Genomes of Flowering Plants: The Family of LTR Retrozymes

In 2005, Hammann’s group reported the existence of two canonical type III HHRs in the genome of the plant *A. thaliana* [57]. Both motifs are located within 5 kb on the plant chromosome 4; one placed at the 3′ end of an open reading frame (antisense) and the second in an intergenic non-protein-coding region. Molecular analysis in various plant tissues confirmed that at least one of the motifs is transcribed and self-cleaves. The in vivo expression and activity of these HHRs, together with their sequence conservation, suggest an unidentified biological function for these motifs in the plant [57]. Later on, bioinformatic searches unveiled the occurrence of numerous type III HHR motifs in the genomes of diverse angiosperms [62]. The ribozymes were mostly found in dicot plants, and following a patchy distribution (i.e., they occur frequently in some plant species, but they are absent in others). The genomic HHRs in plants are usually found as either monomers or dimer repeats separated by a few hundred base pairs (600–1000 bp), although some examples of trimer and even tetramer repeats can be also detected [41]. Comparative analysis of the sequences flanked by the dimeric HHRs indicated that they have almost no identity among different plant species. However, they all show a similar topology, as described in Figure 2. The HHR-containing elements are delimited by a variable 4 bp repeat that matches typical target side duplications (TSDs). The HHR motifs are found embedded in direct repeats of ~350 bp, which can be regarded as long terminal repeats (LTRs). Both LTRs delimit a central non-protein-coding region (~300–700 bp), which begins with the sequence of a primer binding site (PBS) corresponding to the complementary tRNA^Met^ of the plant, and finishes with a poly-purine tract (PPT), which are both characteristic elements of LTR-retrotransposons [76]. Altogether, these elements were classified as a new family of non-autonomous retrotransposons with hammerhead ribozymes or retrozymes, similar to other small and non-autonomous retrotransposons of plants such as the terminal-repeat retrotransposons in miniature (TRIMs) [77] or small LTR retrotransposons (SMARTs) [78]. There are clear similarities between the PBS and the 5′- and 3′- ends of plant retrozymes and those of retrotransposons of the Ty3-Gypsy family, suggesting that these autonomous LTR retrotranposons may carry out the mobilization of the retrozymes [41] (Figure 2).

Our analyses of diverse somatic and reproductive plant tissues revealed a high accumulation (~0.1–1 ng/μg of total plant RNAs) of circular and, to a lesser extent, linear RNAs corresponding to the sequence encompassed by the HHR motifs. These results confirmed that retrozymes are transcribed in most plant tissues, but also that the HHRs self-cleave in vivo, followed by RNA circularization. This step of circularization is efficiently carried out in vitro by a chloroplastic isoform of a plant tRNA ligase [21], in a similar way as described for viroids with HHRs of the family *Avsunviroidae* [79]. Moreover, despite the low sequence identity between most plant retrozymes, their secondary structure predictions result in a similar highly branched RNA architecture, which resembles the structures predicted for some infectious circRNAs of plants such as viroids and viral RNA satellites (Figure 4).

## 6. Diverse Type I HHRs Occur in Autonomous and in Non-autonomous Metazoan Retrotransposons

Type I HHRs (Figure 3) were originally reported in the repetitive DNA (so called satellite DNA) of a few animals, both vertebrates [57] and invertebrates [58,59]. More recent studies extended the occurrence of similar type I HHRs to a large collection of metazoan genomes, from cnidarians to mammals [62,67]. These metazoan ribozymes usually occur as multiple copies in tandem repeats, suggesting that the sequence repeats with type I HHRs in animal genomes constitute a new family of retrozymes similar to the plant LTR retrozymes with type III HHRs. However, plant and metazoan retrozymes show specific differences, such as the size of the sequence repeats encompassed by the HHRs (~150–300 bp in animals vs. ~700 bp in plants), different GC content (below 50% in metazoans), and the absence in metazoan retrozymes of LTRs, PBS or PPT motifs. Moreover, most type I HHRs show a characteristic set of tertiary interactions different to the ones found in type III HHRs, as well as a weak helix III that, in many cases, prevents the self-cleavage as monomeric but not as dimeric motifs (Figure 3) [46,83]. The characterization of these metazoan non-LTR retrozymes from three disparate organisms, such as a cnidarian (the coral *Acropora millepora*), a mollusc (the mussel *Mytilus galloprovincialis*) and an amphibian (the axolotl *Ambystoma mexicanum*), confirmed the accumulation of high levels of circRNAs in all the tissues analyzed [42]. These circRNAs are predicted to adopt stable and highly self-paired secondary structures (Figure 5). Moreover, we confirmed in vitro that most of the type I HHRs in invertebrate genomes are able to adopt a stable helix III and, consequently, to self-cleave efficiently as monomeric motifs. Amphibian motifs, on the other hand, lack a clear helix III, and require dimeric constructs to self-cleave efficiently.

On the other hand, previous bioinformatic searches reported that many metazoan genomes contain thousands of minimal type I HHRs [74], which usually lack any of the non-conserved nucleotides of the helix III (Figure 3). Most of these atypical ribozymes were found as dispersed motifs in non-coding regions of the animal genomes, but in some cases, they mapped to the pseudo-LTRs of Penelope-like retroelements (PLEs) [74]. Analysis of PLE sequences from diverse animal taxa confirmed the conserved occurrence of similar minimal type I HHRs in tandem, which may self-cleave through dimeric motifs [74,84]. More recently, a new family of giant retrotransposons attached to telomeric repeats, the so called Terminons, have been found to contain PLE-like HHRs conserved in diverse regions of their sequence [75]. The in vivo activity and role of these minimal type I HHRs in the biology of these retrotransposons are still a mystery. In any case, the presence of type I HHRs in both PLEs/Terminons and in non-LTR retrozymes, as well as their co-occurrence in all the genomes analyzed, indicate that autonomous PLEs/Terminons retrotransposons are likely candidates to complete the mobilization of non-autonomous non-LTR retrozymes (Figure 2) [42].

## 7. A general Model for Retrozyme Spreading in Plant and Animal Genomes

The family of plant LTR retrozymes is similar to other small non-autonomous LTR-retrotransposons of plants, such as TRIMs [77] and SMARTs [78], whereas metazoan non-LTR retrozymes could be regarded as a novel family of single interspersed nucleotide elements (SINEs). However, their constitutive expression as small circRNAs in all analyzed organisms indicates that they are an exceptional family of eukaryotic retrotransposable elements. As with many non-autonomous retrotransposons, plant and animal retrozymes show no protein-coding regions, but self-cleaving ribozymes responsible for circRNA formation. This RNA motif seems to be a key element in the spreading of retrozymes (Figure 2), which would start with the transcription of the genomic copies by a cellular polymerase. Neither plant nor metazoan retrozymes seem to harbour transcriptional promoters, suggesting that retrozymes may undergo Pol-driven (either I, II or III) read-through transcription depending on tissues and/or their genomic location. Nascent RNA transcripts would follow co-transcriptional self-processing by either monomeric or dimeric HHR ribozymes, producing linear RNAs with 5′-OH and 2′-3′-cyclic-phosphate ends. Finally, the circularization of the self-cleaved RNAs would be carried out by a host RNA ligase factor [41,42]. In the case of plants, the resulting circRNAs could bind cellular tRNA^Met^ through their PBS motifs, which would prime the reverse transcriptases encoded by LTR-retrotransposons of the Ty3-Gypsy family. In the case of metazoan non-LTR retrozymes, the sequence repeats do not show any conserved motif that could help deduce the autonomous retrotransposon whose transpositional machinery they may utilize. In any case, the most likely pathway for both retrozyme families implies a reverse transcriptase that, properly primed and thanks to the circular nature of the RNA template, would produce cDNAs of more than one unit in length. Finally, the resulting cDNAs would be integrated in new genomic locations through the machinery of the autonomous retrotransposons (Figure 2).

A last question to be addressed is related to the high levels of circRNAs with HHRs detected in most organisms analyzed, which suggest that these RNA circles may have other biological functions different from their own mobilization. The discovery of splicing-derived circRNAs involved in diverse forms of gene regulation [40] offers new hints about the potential roles of circRNAs with HHRs present in the cell. On the other hand, genomic retrozymes are frequently found in many copies (depending on the organism, from dozens to thousands of repeats), which suggests that even low transcription activity would result in abundant levels of circRNAs. However, most of the data obtained in plants indicate that only a few retrozyme copies may be transcriptionally active [38], which suggests that the higher stability of these structured circRNAs, as compared to linear RNAs, could be the reason for their high levels of accumulation in vivo. Moreover, the presence of a high sequence heterogeneity in any given population of circRNAs in both plant and metazoan species, together with examples of retrozyme RNAs of the negative polarity, also suggests the intriguing possibility of RNA-to-RNA replication by endogenous polymerases.

## 8. Concluding Remarks

Selfish mobile genetic elements, such as transposons, viruses and other subviral agents, can be found in virtually every cellular life form. These genetic parasites and their interaction with their hosts are a major feature of life evolution [85]. On the one hand, parasite–host coevolution allows a continuous arms race that has resulted in sophisticated pathways of defense and counter-defense. At the same time, these parasites provide an source of new genetic material, which can be domesticated by the host genome to perform new molecular functions [85,86]. In this regard, the heterogeneous group of circRNAs with ribozymes could be considered another player in the evolutionary history of many organisms. Eukaryotic retrozymes are expressed as abundant RNA circles in most tissues, and future research will help us to understand whether these circRNAs and ribozymes play novel biological roles in the cell. Moreover, these widespread genome-encoded circRNAs show clear similarities with infectious circRNAs such as the HDV and some plant virus satellites and viroids: such as small size, highly structured RNAs and the presence of small self-cleaving motifs such as the HHR and HDV ribozymes [21]. This latter characteristic, the presence of small ribozymes, is also found in other autonomous retrotransposons (PLEs, R2, RTEs and other LINEs), which establishes an evolutionary connection between all these elements through these RNA motifs. The recent examples of different HDV-like circRNAs harbouring unrelated catalytic RNAs such as HHRs and HDV ribozymes, indicate that this key feature has a modular nature that may have arisen independently several times during the origin and evolution of these elements. Moreover, similarly to the many known examples of co-option or domestication of sequences from transposable elements by their hosts [86], small ribozymes seem to have followed a similar process. It has been previously reported that the presence of a few highly conserved small ribozymes in vertebrate genomes, such as the mammalian 3′-UTR HHR [61] and diverse intronic HHRs [68] or HDV ribozymes in amniotes [70], has likely been preserved during the evolution of these organisms to perform new and conserved biological functions.

Regarding the origin of this group of circRNAs with ribozymes in eukaryotes, multiple options are possible. We have already proposed a likely scenario where plant LTR retrozymes may have given rise to infectious circRNAs such as viroids and viral satellites with ribozymes [21]. Regarding the origin of genomic LTR retrozymes themselves, one possible explanation could be found in events of horizontal transfer from other organisms containing elements with type III HHRs, such as bacteria or insects [62] (de la Peña, in preparation). Similarly, we can envisage that HDV-like circRNAs could have originated several times during evolution from a eukaryotic RNA encoding a Delta-like antigen, having acquired either HDV or HHR ribozymes from any of the ubiquitous genomic retrotransposons. Altogether, these data strongly suggest that we are just beginning to scratch the surface, and new self-cleaving ribozymes and their circRNAs will be likely discovered in the future.

## Figures and Tables

**Figure 1 cells-09-02555-f001:**
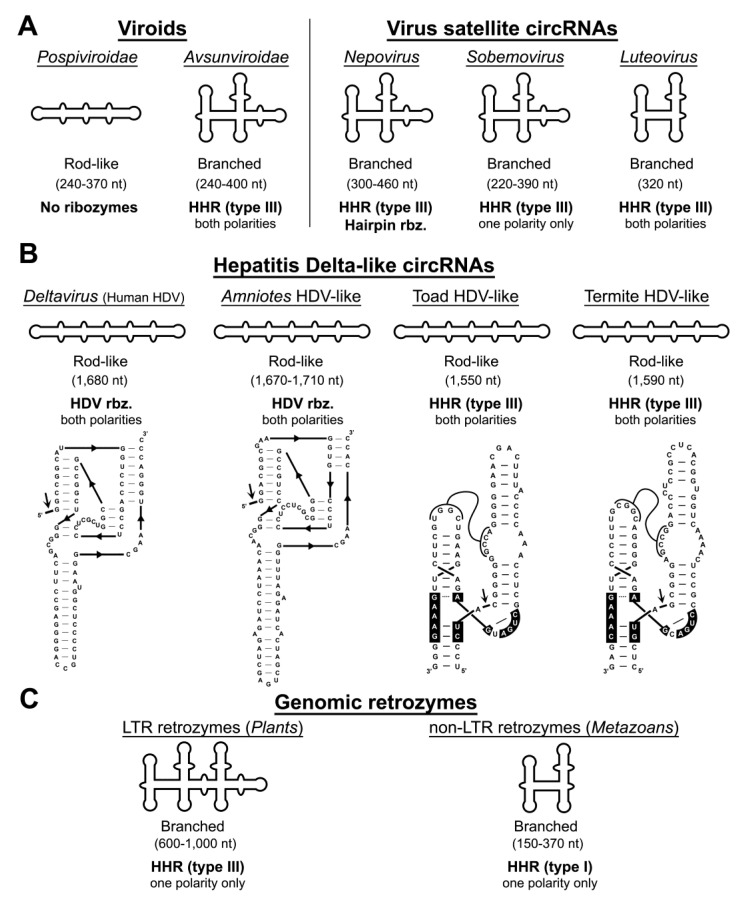
(**A**) Infectious circular RNAs of flowering plants such as viroids (left) and viral satellite circular RNAs (circRNAs) (right). (**B**) Examples of hepatitis delta virus (HDV) circRNAs detected in humans, amniotes, a toad and a termite. An example of the antigenomic ribozyme of each HDV-like circRNA is shown. (**C**) Examples of genome-encoded circRNA retrozymes in plants (left, long terminal repeat (LTR) retrozymes) and animals (rigth, non-LTR retrozymes). The global RNA conformations, typical sizes, and ribozyme classes are indicated for each example.

**Figure 2 cells-09-02555-f002:**
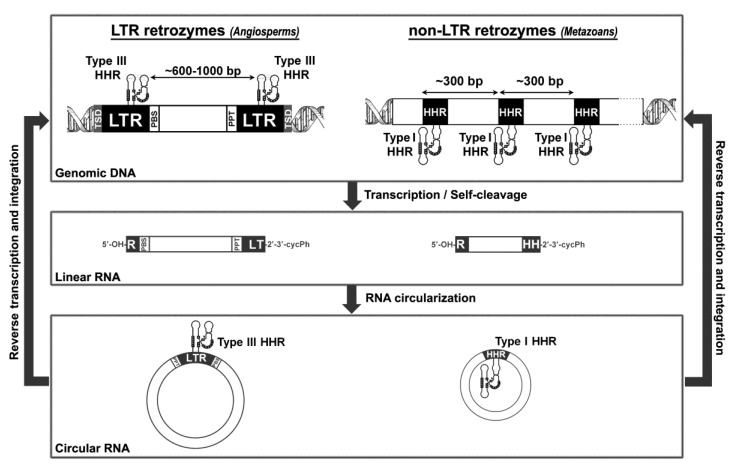
Model for the life cycle of long terminal repeat (LTR) and non-LTR retrozymes. Schematic representation (top) of genomic retrozymes from plants (**left**) and animals (**right**), with their particular features (target site duplications, TSDs; long terminal repeats, LTR; primer binding site, PBS; polypurine tract, PPT). Transcription of these elements followed by hammerhead ribozyme (HHR) self-cleavage results in linear monomeric RNAs with 5′-OH and 2′-3′-cyclic phosphate ends (middle). Monomer circularization (bottom) results in circRNAs that can be reverse-transcribed as multimeric copies due to their circular nature, and resulting cDNAs integrated in new genomic loci.

**Figure 3 cells-09-02555-f003:**
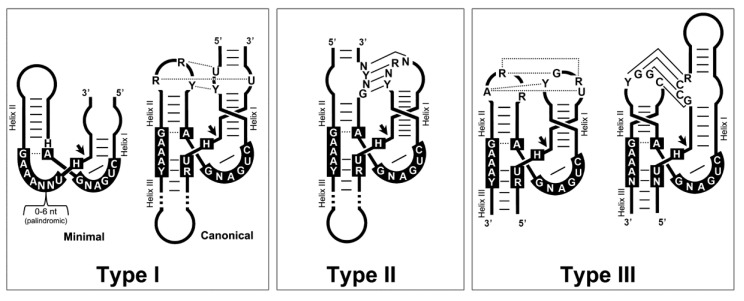
Three-dimensional diagrams of the three possible HHR topologies (Type I, II and III). Black boxes indicate the conserved nucleotides at the catalytic core. Dotted and continuous lines refer to non-canonical and Watson–Crick base pairs, respectively. The three topologies have been reported in the genomes of bacteriophages and prokaryotes. Type I HHRs are mostly found in metazoan genomes, whereas Type III motifs are found in plants and infectious circRNAs, such as viroid-like and some HDV-like agents. N stands for any nucleotide, whereas R stands for purines (A or G), Y for pyrimidines (U or C), and H for either A, U or C.

**Figure 4 cells-09-02555-f004:**
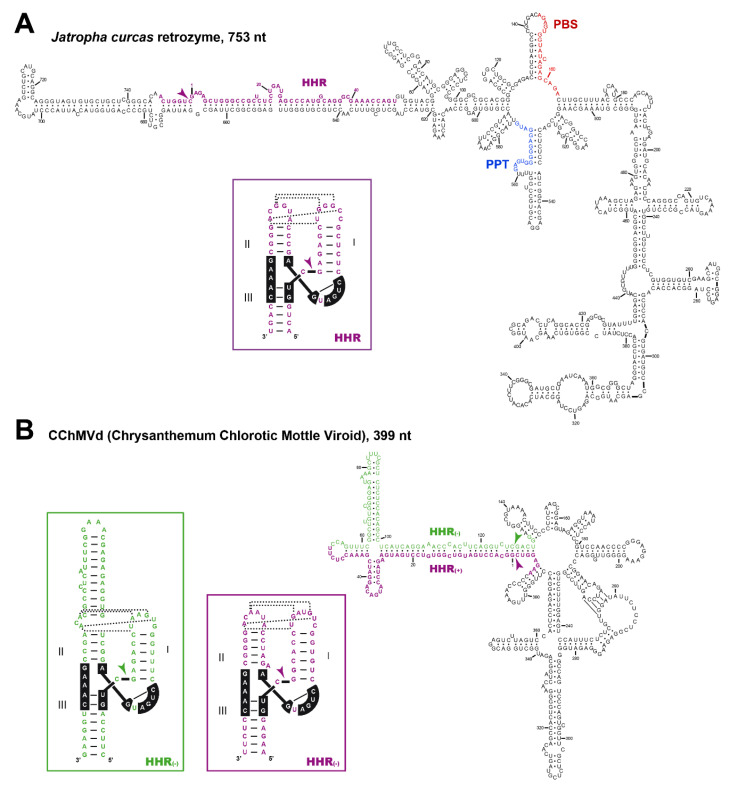
Minimum free energy secondary structure predictions for (**A**) a retrozyme circRNA of *Jatropha curcas* (Entry KX273075.1) and (**B**) the viroid CChMVd (Entry AJ878085.1). HHR sequences are shown in purple (positive polarity) and green (negative polarity). The corresponding structure of the HHRs motifs are shown under each circRNA structure. Dotted lines indicate the predicted tertiary interactions between HHR loops based on previous models [80,81]. Self-cleavage sites are indicated with arrows. Kissing-loop interactions described for CChMVd [82] are shown. Numbering for each circRNA starts at the self-cleavage site.

**Figure 5 cells-09-02555-f005:**
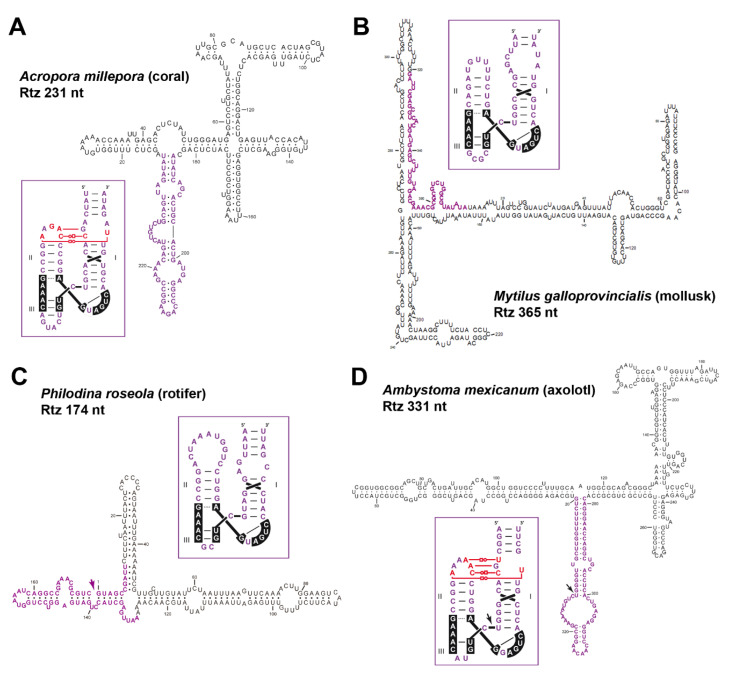
Minimum free energy secondary structure prediction for several examples of circRNAs derived from metazoan non-LTR retrozymes such as (**A**) a coral, (**B**) a mollusk, (**C**) a rotifer, and (**D**) a salamander. HHR sequences are shown in purple letters, with known tertiary interactions drawn in red. The self-cleavage sites are indicated with arrows. Numbering starts at the HHR self-cleavage site.

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
