# Peer review of "A Singular and Widespread Group of Mobile Genetic Elements: RNA Circles with Autocatalytic Ribozymes"

_cells, 2020, doi:10.3390/cells9122555_

Round 1

Reviewer 1 Report

This review describes the state of the art of circular RNAs found across organisms and their associated ribozymes. It is globally well written and complete. The figures are well-done. Clarityand accessibility to a non-specialist audience can be improved following the points below.

General: please add a sentence at the beginning of each section, summarizing the content of it.

L25. The abstract finishes with potential applications which are not discussed in the text. It would be more useful to summarize what is understood at this point and remains to be understood.

L31. “weird” is imprecise, explain.

L47. The title of section 2 should better describe the content than being story like.

L93. “more complex” than what?

L97. Space before “Among”

L100. “at” instead of “in” “the frontiers”

L100. “Discovery as most ancient biocatalyst”: this is a hypothesis, not a discovery

L138. It would be useful to minimally describe the process represented in Fig. 2 in the main text.

L140. Define LTR the first time it appears.

L146. “Autocatalytic” is a word reserved to entities helping the production of copies of themselves, conditional to the fact that they are initially present (‘A makes more A’). “Self-splicing” is more appropriate (‘A makes itself from B’).

L149. One may remove ‘definitely’ and ‘unique’.

L146. “RNA circles”: clarify the relationship between RNA circles and self-cleaving ribozymes. Are the latter always associated with a circRNA or not, and do we even know when it is the case? It is hard to deduce from the text.

L227. A symbol I do not know of in “ng/?g”.

L329. Functional roles are suggested, but it would be good to state that this remains an open question.

L334. ‘establishes’ with a ‘e’

Author Response

This review describes the state of the art of circular RNAs found across organisms and their associated ribozymes. It is globally well written and complete. The figures are well-done. Clarityand accessibility to a non-specialist audience can be improved following the points below.

We would like to thank the reviewer for his/her interest for our manuscript.

General: please add a sentence at the beginning of each section, summarizing the content of it.

L25. The abstract finishes with potential applications which are not discussed in the text. It would be more useful to summarize what is understood at this point and remains to be understood.

As suggested by the referee, we have removed from the abstract the sentence about potential applications, which was not discussed in the text. We have also included a couple of new sentences in the manuscript about the hypothetical biological roles of genome-encoded circRNAs with ribozymes due to their high abundance and their "similarities" with splicing-derived circRNAs.

L31. “weird” is imprecise, explain.

We have changed the imprecise word "weird" by "peculiar". Since their discovery 40 years ago, circRNAs had been regarded as "different to what is normal", "strange" or "peculiar", whereas linear RNAs were considered as the typical form of the RNA macromolecule.

L47. The title of section 2 should better describe the content than being story like.

As suggested by the referee, we have changed the title of section 2 to "The discovery of the first circRNAs"

L93. “more complex” than what?

As suggested, the sentence has been changed to "open a more complex scenario than previously thought for the family of infectious circRNAs"

L97. Space before “Among”

L100. “at” instead of “in” “the frontiers”

We would like to thank the referee for bringing these mistakes to our attention. Typos have been now fixed.

L100. “Discovery as most ancient biocatalyst”: this is a hypothesis, not a discovery

As suggested, the word "discovery" was replaced by "hypothesis"

L138. It would be useful to minimally describe the process represented in Fig. 2 in the main text.

As suggested, we have included in the legend of Figure 2 a minimal description of this process. It is also explained in section 7 of the main text.

L140. Define LTR the first time it appears.

As suggested, the term LTR was defined

L146. “Autocatalytic” is a word reserved to entities helping the production of copies of themselves, conditional to the fact that they are initially present (‘A makes more A’). “Self-splicing” is more appropriate (‘A makes itself from B’).

Formally, "autocatalytic" is used for any chemical reaction where the product is also a catalyst for the same reaction. In the case of a single hammerhead ribozyme, it cannot be chemically considered even a catalyst, because it is consumed in the catalyzed reaction. However, either multimeric ribozyme copies or circRNAs are able to re-generate a single ribozyme after cleavage and circularization, thus the word autocatalytic could be regarded as valid. On the other hand, the expression self-splicing is generally used for the more complex reaction of autocatalytic group I/II introns, which includes cleavage and exon ligation. 

L149. One may remove ‘definitely’ and ‘unique’.

As suggested by two referees, we removed these and other adjectives along the manuscript to make more concise sentences.  

L146. “RNA circles”: clarify the relationship between RNA circles and self-cleaving ribozymes. Are the latter always associated with a circRNA or not, and do we even know when it is the case? It is hard to deduce from the text.

The manuscript is mostly focused on a particular group of circRNAs; those having self-cleaving ribozymes. As pointed in the text, some circRNAs, such as splicing-derived RNA circles or some viroids, do not contain autocatalytic RNAs. To better clarify this point, the title of the section was changed to "Autocatalytic RNA cleavage in diverse RNA circles"

L227. A symbol I do not know of in “ng/?g”.

A mistake due to a font change was fixed 

L329. Functional roles are suggested, but it would be good to state that this remains an open question.

As suggested, we state that future research will help us to understand the possible roles of these circRNAs.

L334. ‘establishes’ with a ‘e’

We would like to thank the referee for bringing this typo to our attention

Reviewer 2 Report

A nice review presenting the state of the art in circRNAs with emphasis on the circRNAs which are studied by the authors. This is an interesting piece for the reader who needs to be updated on the field.

The text could be improved by shortening sentences from place to place.

Author Response

A nice review presenting the state of the art in circRNAs with emphasis on the circRNAs which are studied by the authors. This is an interesting piece for the reader who needs to be updated on the field.

The text could be improved by shortening sentences from place to place.

As suggested by the referee, we have shortened some sentences along the manuscript to make a more concise text

Reviewer 3 Report

-- Comments to the authors:

In the manuscript entitled, “A peculiar and widespread group of mobile genetic elements: RNA circles with autocatalytic ribozymes (cells-993154)”, the authors have a compressive review on the circRNAs harboring or being generated from ribozymes. It is well written and organized. It has an alternative aspect on the biogenesis and functions of circRNAs, and should attract a great attention for scientists working in the related fields. Only a minor problem should be fixed (a few citations was missing).

-- Major points:

> The authors should consider to adding the following literatures for the early evidences regarding circRNAs in mammals (in addition to sry in 1993): Cell 1991, 64(3):607-613 and EMBO J 1992, 11(3):1095-1098.

-- Minor points:

Typing errors: Line 124: ‘syr’ should be ‘sry’ instead.

Author Response

-- Major points:

> The authors should consider to adding the following literatures for the early evidences regarding circRNAs in mammals (in addition to sry in 1993): Cell 1991, 64(3):607-613 and EMBO J 1992, 11(3):1095-1098.

We thank the referee for bringing us to the attention of these two interesting publications. Actually, revisiting the bibliography, we have found out that in January 1993, Cocquerelle et al. (Mis‐splicing yields circular RNA molecules. FASEB J . 1993 Jan;7(1):155-60. doi: 10.1096/fasebj.7.1.7678559) already reported the presence of circRNAs in human cells. We have included all these references to offer a more precise landscape of the history of splicing-derived circRNAs.

-- Minor points:

Typing errors: Line 124: ‘syr’ should be ‘sry’ instead.

We finally decided not to mention this particular gene (see above)